# Phylogenetic and Morphological Evidence for Three New Species of Diaporthales (Ascomycota) from Fujian Province, China

**DOI:** 10.3390/jof10060383

**Published:** 2024-05-27

**Authors:** Taichang Mu, Yongsheng Lin, Nemat O. Keyhani, Huili Pu, Ziying Lv, Chenhui Lan, Jinming Xiong, Xiaohao Chen, Xinyang Zhan, Zhiying Zhao, Huajun Lv, Motunrayo Yemisi Jibola-Shittu, Jianlong Wu, Peisong Jia, Shuaishuai Huang, Junzhi Qiu, Xiayu Guan

**Affiliations:** 1State Key Laboratory of Ecological Pest Control for Fujian and Taiwan Crops, College of Life Sciences, Fujian Agriculture and Forestry University, Fuzhou 350002, China; mutaichang@163.com (T.M.); linyongsheng0909@163.com (Y.L.); hdpuhuili@163.com (H.P.); 18359311723@163.com (Z.L.); 18368654108@163.com (C.L.); serendipity879440255843996@outlook.com (X.C.); 1468550268@139.com (X.Z.); zhaozhiyingv1230@163.com (Z.Z.); 13328626122@163.com (H.L.); motunjibolashittu@gmail.com (M.Y.J.-S.); 2Department of Biological Sciences, University of Illinois, Chicago, IL 60607, USA; keyhani@uic.edu; 3Institute of Microbiology, Chinese Academy of Sciences, Beijing 100101, China; xiongxiaojing1016@gmail.com; 4Xiamen Botanical Garden, Xiamen 361004, China; sega999@163.com; 5Institute of Plant Protection, Xinjiang Academy of Agricultural Sciences, Urumqi 830091, China; jps-fly@163.com; 6School of Ecology and Environment, Tibet University, Lhasa 850014, China; alaxender1989@126.com; 7Key Laboratory of Ministry of Education for Genetics, Breeding and Multiple Utilization of Crops, College of Horticulture, Fujian Agriculture and Forestry University, Fuzhou 350002, China

**Keywords:** multigene phylogeny, phytopathogenic fungi, Pseudoplagiostomataceae, taxonomy

## Abstract

Members of the fungal order Diaporthales are sac fungi that include plant pathogens (the notorious chestnut blight fungus), as well as saprobes and endophytes, and are capable of colonizing a wide variety of substrates in different ecosystems, habitats, and hosts worldwide. However, many Diaporthales species remain unidentified, and various inconsistencies within its taxonomic category remain to be resolved. Here, we aimed to identify and classify new species of Diaporthales by using combined morphological and molecular characterization and coupling this information to expand our current phylogenetic understanding of this order. Fungal samples were obtained from dead branches and diseasedleaves of *Camellia* (Theaceae) and *Castanopsis* (Fagaceae) in Fujian Province, China. Based on morphological characteristics and molecular phylogenetic analyses derived from the combined nucleotide sequences of loci of the internal transcribed spacer regions with the intervening 5.8S nrRNA gene (ITS), the 28S large subunit of nuclear ribosomal RNA gene (LSU), the translation elongation factor 1-α gene (*tef1*), the partial beta-tubulin gene (*tub2*), and partial RNA polymerase II second-largest subunit gene (*rpb2*), three new species of Diaporthales were identified and characterized. They are as follows: *Chrysofolia camelliae* sp. nov., *Dendrostoma castanopsidis* sp. nov., and *Pseudoplagiostoma wuyishanense* sp. nov. They are described and illustrated. This study extends our understanding of species diversity within the Diaporthales.

## 1. Introduction

The fungal order Diaporthales consists of 32 families [1,2], and includes several important plant pathogens, e.g., the chestnut blight fungus, *Cryphonectria parasitica*, the soybean stem canker-causing fungus, *Diaporthe phaseolorum*, and the peach canker and citrus fruit stem-end rot fungi, *Diaporthe amygdali* (*Phomopsis amygdali*) and *Diaporthe citri*, respectively [3]. A number of asexual diaporthalean plant pathogens have also been described including the causative agent of grape bitter rot, *Greeneria uvicola*, and the dogwood anthracnose fungus, *Discula destructiva*. Diaporthales are sac fungi, characterized by ascomata with short or long necks, usually having a central column structure, immersed in host tissue, and unitunicate asci, with a refractive ring at maturity [4,5]. Members of the Diaporthales are widely distributed geographically and have many hosts, inhabiting or colonizing a range of plant, animal, soil, and water substrates and habitats [6,7,8]. Plant-associated members are often endophytic, inhabiting plant tissues, or parasitic, causing disease in plants, with others possibly acting as saprobes [9]. Endophytic associations have been reported between *Diaporthe endophytica* (Diaporthaceae)and leaves of *Schinus terebinthifolius* in Thailand [10]. *Phaeoappendicospora thailandensis* (Phaeoappendicosporaceae) was isolated from dead branch of *Quercus* sp. in Thailand [6], and *Pyrispora castaneae* (Pyrisporaceae) was identifiedas a pathogen or saprobe from *Castanea mollissima* leaves in China [8]. Additional plant pathogens within this order include *Cytospora jilongensis* (Cytosporaceae),which causes stem cankers on *Prunus davidiana*, [11] and *Aurifilum cerciana* (Cryphonectriaceae),responsible for foliage blight on *Terminalia mantaly* [12].

The genus *Chrysofolia* Crous & M.J. Wingf. (Cryphonectriaceae) was erected based on the type species *C. colombiana* by Crous et al. [13]. *Chrysofolia* is mainly characterized by separate-to-aggregated pycnidial conidiomata with globose pycnidia [13,14], conidiophores reduced to conidiogenous cells [15], hyaline, smooth, aggregated, and cylindrical-to-ampulliform conidiogenous cells [13,16], and possessing conidia with hyaline, smooth, guttulate, straight-to-allantoid base with flattened hilum. Species within *Chrysofolia* have been reported to inhabit or colonize plant tissues as endophytes and pathogens [16]. Currently, *Chrysofolia* includes six species: *C. barringtoniae* [16], *C. colombiana* [13], *C. coriariae* [15], *C. fructicola* [17], *C. galloides* [18], and *C. kunmingensis* [14].

*Dendrostoma* X.L. Fan & C.M. Tian (Erythrogloeaceae) was established with the type species *Dendrostoma mali*, with *D. osmanthi* and *D. quercinum* subsequently described by Fan et al. [1]. The asexual morph of *Dendrostoma* is characterized by clavate asci with fusoid-to-cylindrical, bicellular, multiguttulate ascospores [1,19]. Based on morphological characteristics and multi-locus (ITS–LSU–*rpb2*–*tef1*) data, Jiang et al. identified ten new species of *Dendrostoma*: *D. aurorae*, *D. castaneae*, *D. castaneicola*, *D. chinense*, *D. dispersum*, *D. parasiticum*, *D. qinlingense*, *D. quercus*, *D. shaanxiense*, and *D. shandongense* in China [20].

*Pseudoplagiostoma eucalypti* is the type species of *Pseudoplagiostoma* Cheew., M.J. Wingf. & Crous (Pseudoplagiostomataceae) as introduced by Cheewangkoon et al. [21]. Currently, Pseudoplagiostomataceae comprises the single genus of *Pseudoplagiostoma*. Species of *Pseudoplagiostoma* include plant pathogens, endophytes, and saprophytes [21,22,23]. It is worth noting that most of the species *Pseudoplagiostoma* have been found in Asia, indicating the high likelihood of species in this area [23].

Here we identify three new species within the Diaporthales, with *Chrysofolia camelliae* sp. nov. in Cryphonectriaceae, *Dendrostoma castanopsidis* sp. nov. in Erythrogloeaceae, and *Pseudoplagiostoma wuyishanense* sp. nov. in the Pseudoplagiostomataceae of this order. Characterization included a combined morphological and molecular phylogenetic analyses, with the latter including data generated from multi-locus nucleotide sequencing that included the internal transcribed spacer regions with the intervening 5.8S nrRNA gene (ITS), the 28S large subunit of nuclear ribosomal RNA gene (LSU), the translation elongation factor 1-α gene (*tef1*), the partial beta-tubulin gene (*tub2*), and partial RNA polymerase II second-largest subunit gene (*rpb2*). These data establish new members of the Diaporthales, helping improve placement of various genera within this order.

## 2. Materials and Methods

### 2.1. Sample Collection, Isolation, and Morphological Observations

Plant samples of *Camellia*, *Castanopsis fargesii*, and an unidentified tree were collected from Sanming, Quanzhou, and Wuyishan City of Fujian Province, China, in September 2022. Samples were treated as described in Fu et al. [24]. Fungal isolates were single colonies purified by culturing on potato dextrose agar (PDA) in a light incubator at 25 °C and under a 12 h light/dark regime. To promote sporulation, some strains were specially removed to synthetic low nutrient agar (SNA) containing sterilized pine needles [15]. Dried holotype specimens were deposited in the Herbarium Mycologicum Academiae Sinicae, Institute of Microbiology, Chinese Academy of Sciences, Beijing, China (HMAS). Ex-type living cultures were conserved in the China General Microbiological Culture Collection Center (CGMCC). Colony morphologies were captured by camera (Canon EOS 6D MarkII, Tokyo, Japan) at 7 and 14 d after inoculation [23]. Micromorphological characteristics of strains were recorded and observed and photographed using a stereo microscope (Nikon SMZ745, Tokyo, Japan) and biological microscope (Ni-U, Tokyo, Japan) with a digital camera (Olympus, Tokyo, Japan). Conidia of some fungal isolates were observed by a scanning electron microscope (Hitachi TM3030 PLUS, Tokyo, Japan) with measurements of micromorphological structures, carried out using Digimizer 5.4.4 software (https://www.digimizer.com, accessed on 2 April 2024) [25]. All fungal strains were stored in 10% sterilized glycerin and sterile water at 4 °C in 2.0 mL tubes. Taxonomic information of the new taxa were registered in MycoBank (http://www.mycobank.org, accessed on 15 April 2024).

### 2.2. DNA Extraction, PCR Amplification, and Sequencing

Fungal genomic DNA was extracted from growing mycelia using the Fungal DNA Mini Kit (OMEGA-D3390, Feiyang Biological Engineering Co., Ltd., Guangzhou, China) following the manufacturer’s instructions. A total of five gene loci were examined by polymerase chain reaction (PCR) amplification of nucleotide sequences that included regions of ITS, LSU, *tef1*, *tub2*, and *rpb2* genes. The PCR thermal cycle program and primer pairs are listed in Table 1. The PCR reaction volume was 25 µL, containing 12.5 μL of 2 × Rapid Taq Master Mix (Vazyme, Nanjing, China), 1 μL of each forward and reverse primer (10 μM) (Sangon, Shanghai, China), 1 μL of template genomic DNA, and 9.5 µL of double-distilled water (ddH_2_O), and amplification was performed using a Bio-Rad Thermocycler (Hercules, CA, USA). The integrity and sizes of all the PCR products were checked on 1% agarose gel electrophoresis and products were sequenced using a commercial company (Tsingke Co., Ltd., Fuzhou, China). Both forward and reverse sequences of PCR products for each loci were obtained and processed by MEGA 7.0.20 software [26]. New sequences generated in this study have been deposited in GenBank (https://www.ncbi.nlm.nih.gov, accessed on 15 April 2024, Table 2, Table 3 and Table 4).

### 2.3. Phylogenetic Analyses

NCBI-Blast searches using sequence data generated from fungal samples were used to identify and download data from GenBank for multi-locus phylogenetic analyses (Table 2, Table 3 and Table 4). Gene sequences were initially aligned with MAFFTv.7 and optimized manually with MEGA 7.0.20 software and trimAL v1.2 (http://trimal.cgenomics.org, accessed on 3 April 2024) [14,26,33]. Multi-locus phylogenetic analyses of the concatenated aligned dataset were obtained by maximum likelihood (ML) and Bayesian inference (BI) methods. They were inferred using IQtree 1.6.8 [34] and MrBayes 3.2.6 [35] with Phylosuite software v1.2.3 (https://dongzhang0725.github.io/, accessed on 3 April 2024) [36]. IQtree was run under an edge-linked partition model for 5000 ultrafast bootstraps [37]. For Bayesian inference analyses, PartitionFinder2 was used to select the best-fit partition model [38]. The best evolutionary models were used (2 parallel runs, 2,000,000 generations, sample frequency = 100), in which the first 1/4 of trees were discarded as burn-in and the remaining 3/4 of trees were used to calculate Bayesian Posterior Probabilities (BYPP). The phylogenetic trees were visualized using FigTree1.4.4 (http://tree.bio.ed.ac.uk/software/figtree, accessed on 3 April 2024) and embellished with Adobe Illustrator CS 6.0 (Adobe Systems Inc., San Jose, CA, USA).

## 3. Results

### 3.1. Phylogenetic Analyses

For construction of the phylogenetic tree of isolates matching *Chrysofolia*, the concatenated sequence dataset of ITS, LSU, *tef1*, and *tub2* was used which included 27 taxa with *Dwiroopa punicae* (CBS 143163) as the outgroup (Figure 1). The aligned four-locus datasets had an aligned length of 2633 total characters (i.e., ITS: 1–674, LSU: 675–1517, *tef1*: 1518–2032, *tub2*: 2033–2633) including gaps, of which 1906 characters were constant, 253 were variable and parsimony-uninformative, and 474 were parsimony-informative. The best model for the dataset was estimated by PartitionFinder2, and selected in the Bayesian analysis were SYM + I + G for ITS (Lset nst = 6, rates = invgamma), GTR + I + G for LSU (Lset nst = 6, rates = invgamma), HKY + G for *tef1* (Lset nst = 2, rates = gamma), and GTR + G for *tub2* (Lset nst = 6, rates = gamma). Bayesian analyses resulted in an average standard deviation of split frequencies = 0.004399. The topology of the ML tree was similar to Bayesian analyses; thus, the Bayesian tree is shown. Maximum-likelihood bootstrap support (ML-BS) values (≥80%) and Bayesian posterior probabilities (BI-PP) values (≥0.90) are provided as the first and second positions, respectively, above the nodes.

For construction of the phylogenetic tree for analysis of the *Dendrostoma*-matching isolates, the concatenated sequence dataset of ITS, LSU, *rpb2*, and *tef1* was analyzed using sequences from 42 taxa with *Disculoides eucalypti* (CBS 132183) as an outgroup (Figure 2). The aligned four-locus datasets had an aligned length of 2804 total characters (i.e., ITS: 1–503, LSU: 504–1344, *rpb2*: 1345–2419, *tef1*: 2420–2804) including gaps, of which 2208 characters were constant, 137 were variable and parsimony-uninformative, and 459 were parsimony-informative. The best model for the dataset was estimated by PartitionFinder2, and selected in the Bayesian analysis were SYM + I + G for ITS (Lset nst = 6, rates = invgamma), GTR + I + G for LSU and *rpb2* (Lset nst = 6, rates = invgamma), and GTR + G for *tef1* (Lset nst = 6, rates = gamma). Bayesian analyses resulted in an average standard deviation of split frequencies = 0.002505. The topology of the ML tree was similar to Bayesian analyses; thus, the Bayesian tree is displayed. Maximum-likelihood bootstrap support (ML-BS) values (≥80%) and Bayesian posterior probabilities (BI-PP) values (≥0.90) are provided as the first and second positions, respectively, above the nodes.

For construction of the phylogenetic tree for analysis of the *Pseudoplagiostoma*-matching isolates, the concatenated sequence dataset of ITS, LSU, *rpb2*, *tef1*, and *tub2* was analyzed using 22 taxa with *Apoharknessia insueta* (CBS 111377) as the outgroup (Figure 3). The aligned five-locus datasets had an aligned length of 3060 total characters (i.e., ITS: 1–649, LSU: 650–1505, *rpb2*: 1506–2081, *tef1*: 2082–2570, *tub2*: 2571–3060) including gaps, of which 1984 characters were constant, 169 were variable and parsimony-uninformative, and 907 were parsimony-informative. The best model for the dataset was estimated by PartitionFinder2, and selected in the Bayesian analysis were GTR + I + G for ITS, LSU and *tef1* (Lset nst = 6, rates = invgamma), and GTR + G for *rpb2* and *tub2* (Lset nst = 6, rates = gamma). Bayesian analyses resulted in an average standard deviation of split frequencies = 0.000884. The Maximum-likelihood resulted in a similar topology to Bayesian inference; thus, the Bayesian tree is provided. Maximum-likelihood bootstrap support (ML-BS) values (≥80%) and Bayesian posterior probabilities (BI-PP) values (≥0.90) are shown as the first and second positions, respectively, above the nodes.

### 3.2. Taxonomy

#### 3.2.1. *Chrysofolia camelliae* T.C. Mu and J.Z. Qiu, sp. nov., Figure 4

MycoBank Number: MB853510

Etymology: the epithet “*camelliae*” refers to the host genus, *Camellia*.

Holotype: China: Fujian Province, Sanming City, 26°39′28″ N, 117°51′16″ E, on diseased leaves of *Camellia* sp. (Theaceae), 9 September 2022, holotype HMAS 352951; ex-holotype living culture CGMCC3.27473.

Description: Leaf spots irregular and fawn or umber. Asexual morphs developed on PDA. Conidiomata pycnidial, separated to aggregated, globose to subglobose, orange conidial droplets exuded from ostioles. Conidiophores reduced to conidiogenous cells. Conidiophores cells hyaline, smooth, cylindrical to ampulliform, 4.0–6.0 μm × 1.0–4.0 μm. Conidia hyaline, smooth, aseptate, subfusoid, elongate ellipsoidal, 5.0–9.0 μm × 2.0–4.0 μm, mean = 8.0 μm × 3.0 μm, L/W ratio = 2.0, n = 30. Sexual morph not observed.

Culture characteristics: Colonies on PDA were flat with irregular margin, aerial mycelium orange in the center and white at the edge. Surface white initially, becoming orange with age, reverse golden-orange. Growth on PDA attained 57.3–64.7 mm in diameter after 1 week at 25 °C, growth rate 8.2–9.2 mm/day. Growth on PDA attained 78.7–81.3 mm in diameter after 2 weeks at 25 °C, growth rate 5.6–5.8 mm/day. 

Material examined: China: Fujian Province, Sanming City, 26°39′28″ N, 117°51′16″ E, on diseased leaves of *Camellia* sp. (Theaceae), 9 September 2022, paratype HMAS 352952; ex-paratype living culture CGMCC3.27474.

**Figure 4 jof-10-00383-f004:**
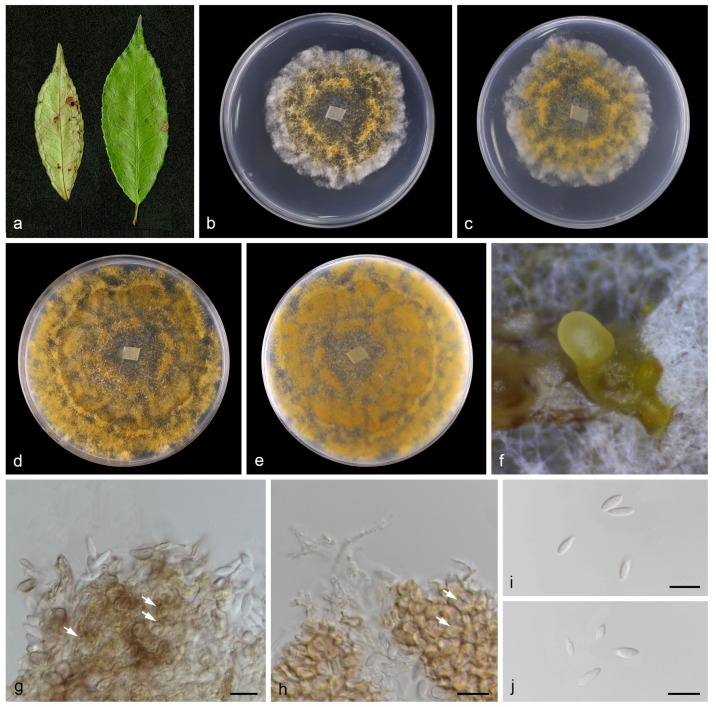
*Chrysofolia camelliae* (holotype HMAS 352951). (**a**) Diseased leaves of *Camellia* sp.; (**b**,**c**) surface and reverse sides of colony after 7 days on PDA (**d**,**e**) and 14 days; (**f**) mass of conidia; (**g**,**h**) conidiogenous cells and conidia; (**i**,**j**) conidia. Scale bars: (**g**–**j**) 10 µm.

Notes: In the current study, to the best of our knowledge, this is the first instance of *Chrysofolia* on leaves of *Camellia* sp. in China, and *Chrysofolia camelliae* sp. nov. is described. Based on the multi-gene phylogenetic analyses of ITS, LSU, *tef1*, and *tub2*, the new species is strongly supported (100% ML/1 PP, Figure 4) with *C. barringtoniae*, *C. colombiana*, *C. coriariae*, *C. fructicola*, *C. galloides*, and *C. kunmingensis*. BLASTn analysis of *C. camelliae* (CGMCC3.27473) and *C. barringtoniae* (TBRC 5647) revealed 28 bp (28/608 bp, 4.6%) nucleotide differences in ITS. BLASTn analysis of *C. camelliae* (CGMCC3.27473) and *C. colombiana* (CPC 24986) revealed 84 bp (84/461 bp, 18.3%) nucleotide differences in *tef1*. BLASTn analysis of *C. camelliae* (CGMCC3.27473) and *C. coriariae* (GUCC 416.4) revealed 81 bp (81/320 bp, 25.3%) nucleotide differences in *tef1* and 55 bp (55/527 bp, 10.4%) nucleotide differences in *tub2*. BLASTn analysis of *C. camelliae* (CGMCC3.27473) and *C. fructicola* (GUCC 194121.1) revealed 60 bp (60/609 bp, 9.9%) nucleotide differences in ITS and 118 bp (118/492 bp, 24.0%) nucleotide differences in *tef1*. BLASTn analysis of *C. camelliae* (CGMCC3.27473) and *C. galloides* (IFRDCC1024) revealed 85 bp (85/466 bp, 18.2%) nucleotide differences in *tef1* and 54 bp (54/523 bp, 10.3%) nucleotide differences in *tub2*. BLASTn analysis of *C. camelliae* (CGMCC3.27473) and *C. kunmingensis* (KUNCC23-13350) revealed 51 bp (51/434 bp, 11.8%) nucleotide differences in *tub2*. Morphologically, the conidia of *C. camelliae* are large than *C. barringtoniae* (5.0–9.0 μm × 2.0–4.0 μm vs. 3.0–7.5 μm × 2.0–3.0 μm). Therefore, we describe this fungus as a new species.

#### 3.2.2. *Dendrostoma castanopsidis* T.C. Mu and J.Z. Qiu, sp. nov., Figure 5

MycoBank Number: MB853511

Etymology: the epithet “*castanopsidis*” refers to the host genus, *Castanopsis*.

**Figure 5 jof-10-00383-f005:**
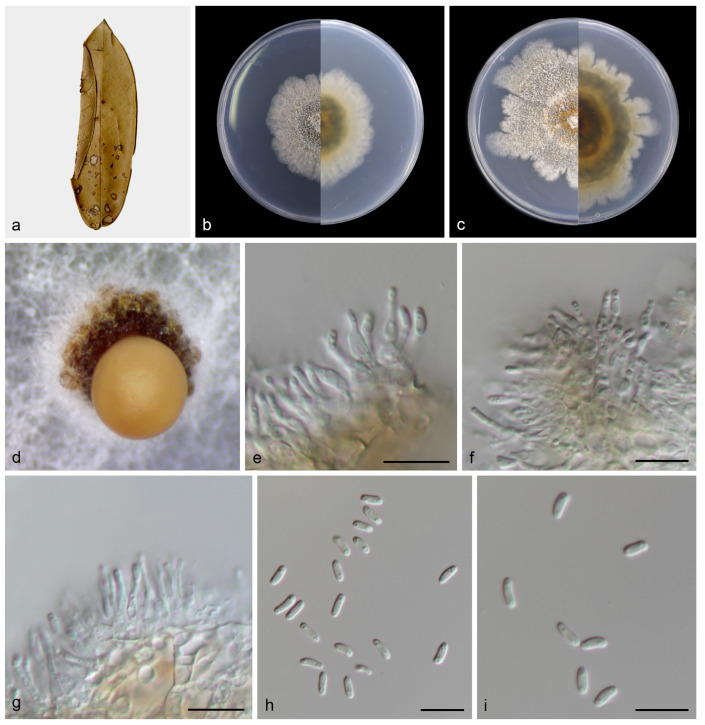
*Dendrostoma castanopsidis* (holotype HMAS 352711). (**a**) Diseased leaves of *Castanopsis fargesii*; (**b**) surface and reverse sides of colony after 7 days on PDA (**c**) and 14 days; (**d**) mass of conidia; (**e**–**g**) conidiogenous cells with conidia; (**h**,**i**) conidia. Scale bars: (**e**–**i**) 10 µm.

Holotype: China: Fujian Province, Quanzhou City, 24°54′38″ N, 117°13′46″ E, on diseased leaves of *Castanopsis fargesii* (Fagaceae), 17 September 2022, holotype HMAS 352711; ex-holotype living culture CGMCC3.25676.

Description: Leaf spots subcircular and irregular, gray to brown. Asexual morphs developed on PDA. Conidiomata pycnidial, separated, conical, orange conidial droplets exuded from ostioles. Conidiophores reduced to conidiogenous cells. Conidiophores cells hyaline, smooth, cylindrical, subcylindrical to ampulliform, 5.0–12.0 μm × 1.0–2.0 μm. Conidia hyaline, smooth, aseptate, biguttulate, cylindric-clavate, 4.0–6.0 μm × 1.0–2.0 μm, mean = 5.0 μm × 2.0 μm, L/W ratio = 3.0, n = 30. Sexual morph not observed.

Culture characteristics: Colonies on PDA were flat and fluffy with pale gray aerial mycelium. Surface white and pale gray initially, then becoming gray and fawnby age, reverse brown. PDA attaining 47.1–48.6 mm in diameter after 1 week at 25 °C, growth rate 6.7–6.9 mm/day. PDA attaining 53.9–72.3 mm in diameter after 2 weeks at 25 °C, growth rate 3.9–5.2 mm/day. 

Material examined: China: Fujian Province, Quanzhou City, 24°54′38″ N, 117°13′46″ E, on diseased leaves of *Castanopsis fargesii* (Fagaceae), 17 September 2022, paratype HMAS 352710; ex-paratype living culture CGMCC3.25675.

Notes: To the best of our knowledge this is the first description of *Castanopsis* on leaves of *Castanopsis fargesii* in China, and *Dendrostoma castanopsidis* sp. nov. is depicted. Based on the multigene phylogenetic analyses of ITS, LSU, *tef1*, and *rpb2*, our two strains were closer to *D. creticum*, *D. elaeocarpi*, and *D. istriacum* with medium support in their clade (82% ML/0.93 PP, Figure 5). BLASTn analysis of *D. castanopsidis* (CGMCC3.25676) and *D. creticum* (CBS 145802) revealed 33 bp (33/615 bp, 5.4%) nucleotide differences in ITS, 78 bp (78/203 bp, 38.4%) nucleotide differences in *tef1*, and 44 bp (44/912 bp, 4.9%) nucleotide differences in *rpb2*. BLASTn analysis of *D. castanopsidis* (CGMCC3.25676) and *D. elaeocarpi* (CFCC 53113) revealed 31 bp (31/466 bp, 6.7%) nucleotide differences in ITS, 84 bp (84/209 bp, 40.2%) nucleotide differences in *tef1*, and 56 bp (56/711 bp, 7.9%) nucleotide differences in *rpb2*. BLASTn analysis of *D. castanopsidis* (CGMCC3.25676) and *D. istriacum* (CBS 145801) revealed 31 bp (31/616 bp, 5.0%) nucleotide differences in ITS, 90 bp (90/212 bp, 42.5%) nucleotide differences in *tef1*, and 49 bp (49/942 bp, 5.2%) nucleotide differences in *rpb2*. In morphology, the conidia and conidiogenous cells of *D. castanopsidis* are narrower than *D. istriacum* (1.0–2.0 vs. 1.9–2.7 μm; 1.0–2.0 vs. 1.8–5.3 μm). Therefore, we introduce the new fungal species: *Dendrostoma castanopsidis* sp. nov.

#### 3.2.3. *Pseudoplagiostoma wuyishanense* T.C. Mu and J.Z. Qiu, sp. nov., Figure 6

MycoBank Number: MB853512

Etymology: the epithet “*wuyishanense*” refers to the locality, Wuyi Mountain National Nature Reserve.

Holotype: China: Fujian Province, Wuyi Mountain National Nature Reserve, 27°40′13″ N, 117°39′23″ E, on dead branches of an unidentified tree, 6 September 2022, holotype HMAS 352519; ex-holotype living culture CGMCC3.25368.

Description: Saprobic on dead branches of broadleaf tree. Asexual morphs developed on PDA. Conidiomata were pycnidial, globose to subglobose, superficial in culture, black orbrown, solitary or aggregated, exuding cream to pale luteous conidial masses from ostioles. Conidiophores reduced to conidiogenous cells. Conidiophores cells hyaline, smooth, cylindrical to clavate, phialidic, 12.0–18.0 μm × 3.0–4.0 μm. Conidia hyaline, smooth, aseptate, long-oval, subcylindrical, 20.0–26.0 μm × 10.0–14.0 μm, mean = 23.0 μm × 12.0 μm, L/W ratio = 1.8, n = 30, base with inconspicuous to conspicuous hilum (1.0–2.0 µm diam). Sexual morph not observed.

Culture characteristics: Colonies on PDA flat and fluffy with gray aerial mycelium. Surface white and pale gray initially, then becoming gray by age, reverse brown to black in the center and gray margin. Colonies on SNA (containing pine needle) wavy hyphae, aerial mycelium white to grayish. PDA attaining 25.1–29.5 mm in diameter after 1 week at 25 °C, growth rate 3.6–4.2 mm/day. PDA attaining 60.2–67.5 mm in diameter after 2 weeks at 25 °C, growth rate 4.3–4.8 mm/day. SNA attaining 58.7–65.5 mm in diameter after 2 weeks at 25 °C, growth rate 4.2–4.7 mm/day.

**Figure 6 jof-10-00383-f006:**
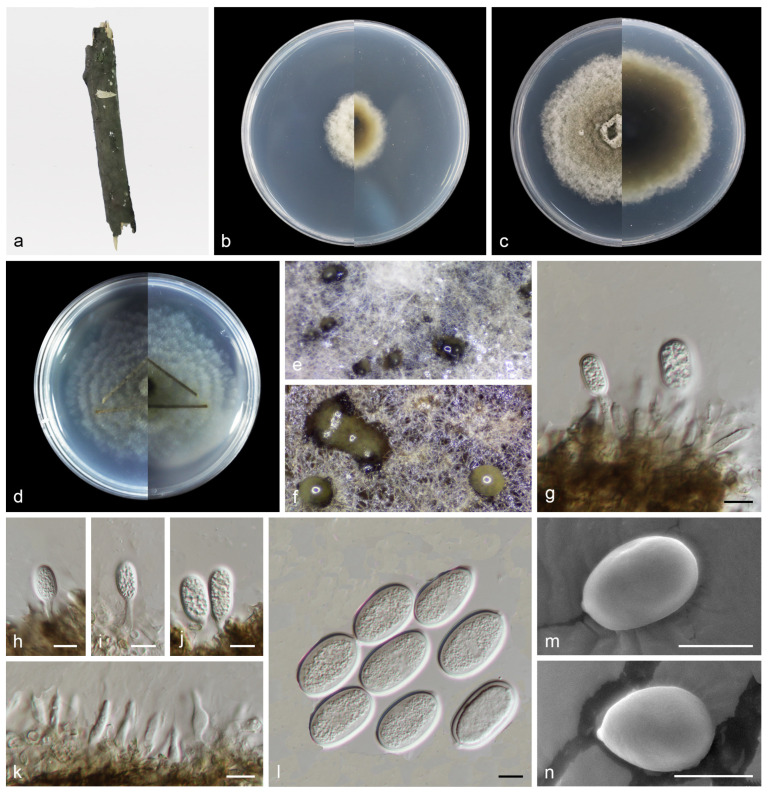
*Pseudoplagiostoma wuyishanense* (holotype HMAS 352519). (**a**) Dead branches; (**b**) surface and reverse sides of colony after 7 days on PDA (**c**) and 14 days; (**d**) surface and reverse sides of colony after 14 days on SNA; (**e**,**f**) mass of conidia; (**g**–**j**) conidiogenous cells with conidia; (**k**) conidiogenous cells; (**l**–**n**) conidia. Scale bars: (**g**–**n**) 10 µm.

Material examined: China: Fujian Province, Wuyi Mountain National Nature Reserve, 27°40′13″ N, 117°39′23″ E, on dead branches of an unidentified tree, 6 September 2022, paratype HMAS 352518; ex-paratype living culture CGMCC3.25367.

Notes: In this study, *Pseudoplagiostoma wuyishanense* sp. nov. is described from dead branches of an unidentified broadleaf tree in China. Tang et al. found *P. dipterocarpicola* from dead twigs of *Dipterocarpus* sp. in northern Thailand [39]. Based on the multigene phylogenetic analyses of ITS, LSU, *tub2*, *tef1*, and *rpb2*, the new species is well supported (100% ML/1 PP, Figure 6) within *Pseudoplagiostoma* and close to *P. bambusae*. *P. bambusae* was described from the diseased leaves of Bambusoideae sp. in China, Fujian Province, Fujian Wuyi Mountain National Nature Reserve [23]. However, BLASTn analysis of *P. wuyishanense* (CGMCC3.25368) and *P. bambusae* (SAUCC 1206-4) revealed 29 bp (29/600 bp, 4.8%) nucleotide differences in ITS, 194 bp (194/835 bp, 23.2%) nucleotide differences in LSU, 63 bp (63/188 bp, 33.5%) nucleotide differences in *tef1*, and 46 bp (46/474 bp, 9.7%) nucleotide differences in *tub2*. Morphologically, the conidia and conidiogenous cells of *P. wuyishanense* are large than *P. bambusae* (20.0–26.0 μm × 10.0–14.0 μm vs. 13–20 μm × 5.7–7.6 μm; 12.0–18.0 μm × 3.0–4.0 μm vs. 5.0–13.0 μm × 1.5–2.5 μm). Therefore, we describe this fungus as a new species. Two species of *Pseudoplagiostoma* (*P. bambusae*, *P. wuyishanense*) were found in the same place (Fujian Wuyi Mountain National Nature Reserve, Fujian Province, China). This indicates that there may be a high species diversity of *Pseudoplagiostoma* in this area.

## 4. Discussion

The Diaporthales constitute a cosmopolitan, highly diverse group of sac fungi that includes saprophytic, endophytic, and phytopathogenic members; however, important aspects of their interfamilial taxonomic relationships remain poorly characterized. A number of Diaporthales species were resolved based on morphological characteristics and multi-locus phylogenetic analyses [15]. In this study, three new species (*Chrysofolia camelliae* sp. nov., *Dendrostoma castanopsidis* sp. nov., and *Pseudoplagiostoma wuyishanense* sp. nov.) belonging to three families (Cryphonectriaceae, Erythrogloeaceae, and Pseudoplagiostomataceae, respectively) in Diaporthales from China were identified and described.

Previously, the identification of genera and species of Diaporthales depended largely on the host specificity and morphology, which led to several incorrect placements (and naming) [10]. However, with the development of molecular phylogenic approaches, more recent comprehensive methods have been applied to resolving issues within the taxonomy of Diaporthales. For example, Cytosporaceae, Diaporthaceae, Gnomoniaceae, and Melanconidaceae were accepted in Diaporthales by phylogenetic analyses of LSU sequences in 2002 [4]. Cryphonectriaceae was established to accommodate *Cryphonectria*, *Chrysoporthe*, *Endothia*, and allied genera by using ITS, LSU, *tub1*, and *tub2* sequence data in 2006 [40]. Cheewangkoon et al. introduced Pseudoplagiostomaceae by using LSU in 2010 [21]. Based on phylogenetic analyses of combined loci of ITS, LSU, *tef1*, and *rpb2*, Senanayake et al. introduced Auratiopycnidiellaceae, Coryneaceae, and Erythrogloeaceae [6]. Recently, Wijayawardene et al. accepted 32 families in Diaporthales [2]. Combined, these advances suggest increased characterization and diversity within Diaporthales. 

Presently, six species of *Chrysofolia* are accepted and have been reported as plant pathogens and endophytes. *Chrysofolia barringtoniae*, *Chrysofolia fructicola*, and *Chrysofolia kunmingensis* were isolated as endophytes from leaves of *Barringtonia acutangula*, *Rosa roxburghii*, and *Coriaria nepalensis*, respectively, whereas *Chrysofolia coriariae* appears to be a pathogen of the *Coriaria nepalensis* plant [14,16,17,18]. Thus, the same species can play multiple roles and may develop into a plant pathogen from initial endophytic interactions under some conditions, e.g., stress or changes to host plant health [41,42].

Twenty-two species of *Dendrostoma* have been reported as tree pathogens, with *Dendrostoma* infecting branches, twigs, and stems of Fagaceae plants, including beeches, chestnuts, and oaks, resulting in production losses important in both agriculture and forestry [20]. The identification of *Dendrostoma castanopsidis* in this study indicated its associated with the evergreen *Castanopsis fargesii* (Fagaceae). Interestingly, Fagaceae is an important flowering plant family (~930 species) with temperate members, mainly deciduous, whereas tropical varieties are evergreens and shrubs [43,44]. With increased collection, fungal Fagaceae plant pathogens, particularly those of *Dendrostoma*,are likely to be found in China and beyond. Although this is the first report on the description of *Dendrostoma castanopsidis* from *Castanopsis fargesii* (Fagaceae), this finding is in line with an earlier study [19] which reported *Dendrostoma* collected from Fagaceae plants, namely *Castanea mollissima* (harboring *D. aurorae*, *D. castaneae*, *D. castaneicola*, *D. chinense*, *D. shaanxiense*, and *D. shandongense*), *C. sativa* (*D. atlanticum*, *D. castaneum*, and *D. luteum*), *Fagus sylvatica* (*D. covidicola*), *Quercus acutissima* (*D. quercinum*), *Q. aliena* (*D. parasiticum* and *D. qinlingense*), *Q. coccifera* (*D. creticum*), *Q. ilex* (*D. istriacum*), *Q. mongolica* (*D. donglingense*), *Quercus* sp. (*D. dispersum*, *D. leiphaemia*, and *D.quercus*), and *Q. wutaishanica* (*D. parasiticum* and *D. qinlingense*).

The identification of a new species of *Pseudoplagiostoma*, *Pseudoplagiostoma wuyishanense* sp. nov., from dead branches reveals that members of the *Pseudoplagiostoma* could potentially contribute to key ecological functions in plant biomass turnover as saprobes [39]. Several studies have reported the abundance of *Pseudoplagiostoma* in Asia. For example, *P. eucalypti* was found from leaf spots in Viet Nam [21]. *P. dipterocarpi* and *P. dipterocarpicola* were also collected from *Dipterocarpus* in Thailand [22,39]. Phookamsak et al. found *P. mangiferae* from leaves of *Mangifera* in China [45]. Zhang et al. described three species of *Pseudoplagiostoma* in China, viz., *P. alsophilae*, *P. bambusae*, and *P. machili*. Through the study and analysis historical biogeography, Asia is probably the ancestral area of *Pseudoplagiostoma* [23]. Despite this, additional studies of *Pseudoplagiostoma* are required to reveal species and ecological diversity.

## 5. Conclusions

In this study, we have identified three new species of Diaporthales isolated in Fujian Province, China, namely *Chrysofolia camelliae* sp. nov. (Cryphonectriaceae), *Dendrostoma castanopsidis* sp. nov. (Erythrogloeaceae), and *Pseudoplagiostoma wuyishanense* sp. nov. (Pseudoplagiostomataceae). Identification was based on morphological and multi-locus phylogenetic analyses, which strongly supported their designations. Our data contribute to the characterization of the ecological diversity of Diaporthales. Identifying the attributes and functional consequences of these fungi is warranted.

## Figures and Tables

**Figure 1 jof-10-00383-f001:**
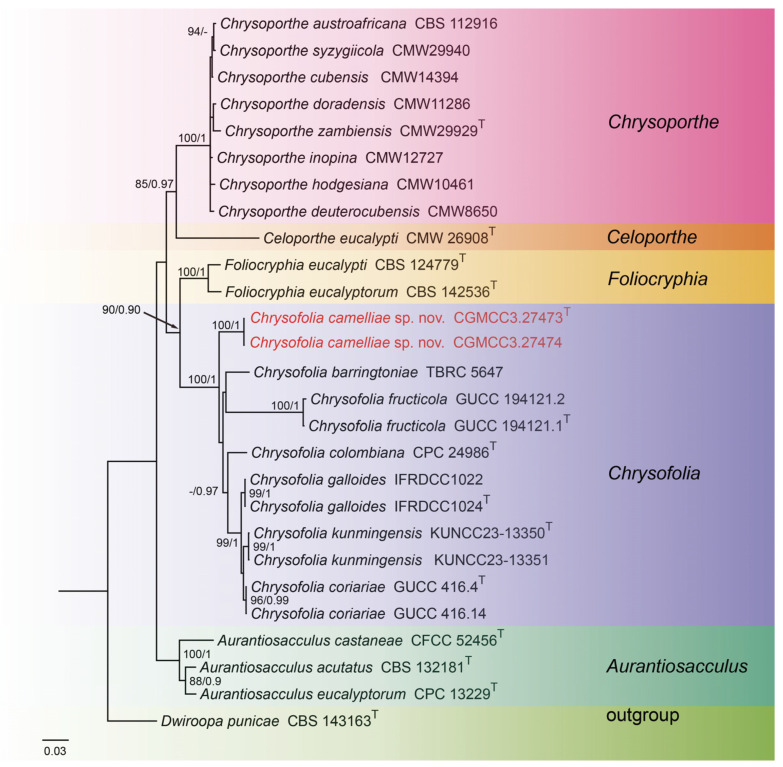
Phylogenetic tree of *Chrysofolia* and allied genera inferred from Bayesian inference analyses based on a combined ITS, LSU, *tef1*, and *tub2* sequence dataset, with *Dwiroopa punicae* (CBS 143163) as outgroup. The ML and BI bootstrap support values above 80% and 0.90 BYPP are indicated at the nodes. Strains marked with “T” are ex-type, ex-epitype, or ex-neotype. The strains from this study are indicated in red.

**Figure 2 jof-10-00383-f002:**
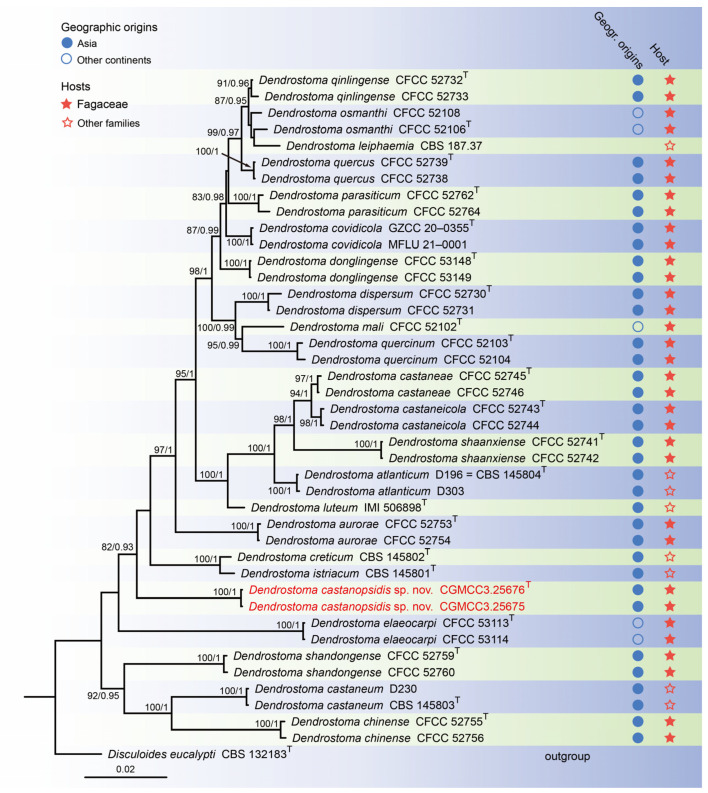
Phylogenetic tree of *Dendrostoma* inferred from Bayesian inference analyses based on a combined ITS, LSU, *rpb2*, and *tef1* sequence dataset, with *Disculoides eucalypti* (CBS 132183) as outgroup. The ML and BI bootstrap support values above 80% and 0.90 BYPP are shown at the nodes. Strains marked with “T” are ex-type, ex-epitype, or ex-neotype. Thestrains from this study are indicated in red.

**Figure 3 jof-10-00383-f003:**
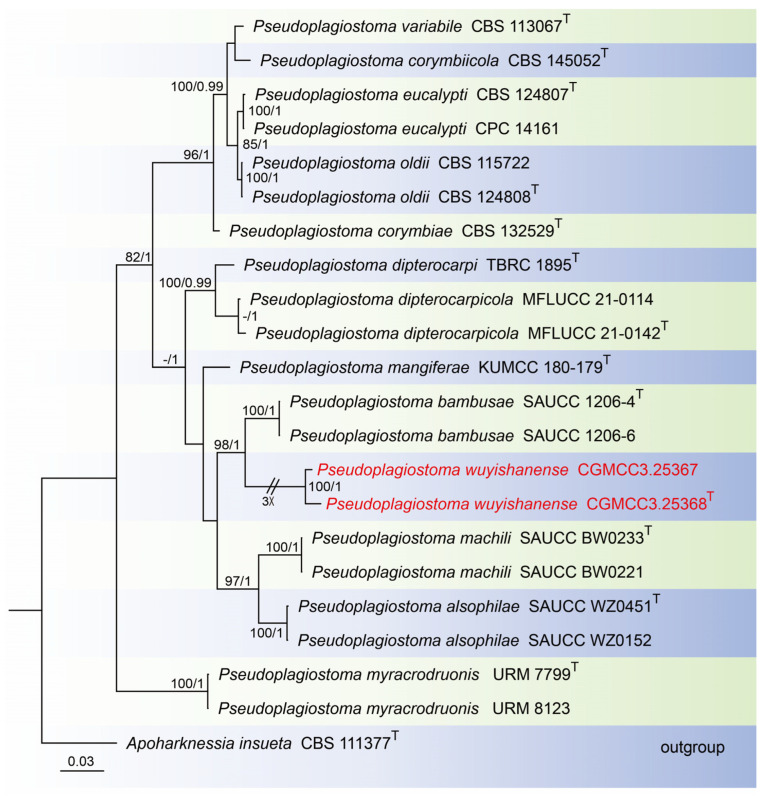
Phylogenetic tree of *Pseudoplagiostoma* inferred from Bayesian inference analyses based on a combined ITS, LSU, *rpb2*, *tef1*, and *tub2* sequence dataset, with *Apoharknessia insueta* (CBS 111377) as outgroup. The ML and BI bootstrap support values above 80% and 0.90 BYPP are shown at the nodes. Strains marked with “T” are ex-type, ex-epitype, or ex-neotype. The strains from this study are indicated in red.

**Table 1 jof-10-00383-t001:** The primer pairs, sequences, and programs in this study.

Locus	Primers	Sequence (5′-3′)	PCR Cycles	References
ITS	ITS5	GGA AGT AAA AGT CGT AAC AAG G	(95 °C: 30 s, 55 °C: 30 s, 72 °C: 1 min) × 35 cycles	[27]
	ITS4	TCC TCC GCT TAT TGA TAT GC		
LSU	LROR	GTA CCC GCT GAA CTT AAG C	(95 °C: 30 s, 52 °C: 30 s, 72 °C: 1 min) × 35 cycles	[28,29]
	LR5	TCC TGA GGG AAA CTT CG		
*rpb2*	fRPB2-5F	GAY GAY MGW GAT CAY TTY GG	(95 °C: 30 s, 56 °C: 30 s, 72 °C: 1 min) × 35 cycles	[30]
	fRPB2-7cR	CCC ATW GCY TGC TTM CCC AT		
*tef1*	EF1-728F	CATCGAGAAGTTCGAGAAGG	(95 °C: 30 s, 56/52 °C: 30 s, 72 °C: 1 min) × 35 cycles	[31]
	TEF1-986R	TAC TTG AAG GAA CCC TTA CC		
*tub2*	Bt2a	GGT AAC CAA ATC GGT GCT GCT TTC	(95 °C: 30 s, 53 °C: 30 s, 72 °C: 1 min) × 35 cycles	[32]
	Bt2b	ACC CTC AGT GTA GTG ACC CTT GGC		

**Table 2 jof-10-00383-t002:** Species names, voucher or culture codes, hosts or substrate, locations, and corresponding GenBank accession numbers of DNA sequences used in the molecular phylogenetic analyses of *Chrysofolia* and allied genera.

Species	Culture/Voucher	Host/Substrate	Locations	GenBank Accession Number			
				ITS	LSU	*tef1*	*tub2*
*Aurantiosacculus acutatus*	CBS 132181 *	*Eucalyptus viminalis*	Australia	JQ685514	JQ685520	MN271823	–
*Aurantiosacculus castaneae*	CFCC 52456 *	*Castanea mollissima*	China	MH514025	MH514015	–	MH539688
*Aurantiosacculus eucalyptorum*	CPC 13229 *	*Eucalyptus globulus*	Australia	JQ685515	JQ685521	MN271824	–
*Celoporthe eucalypti*	CMW 26908*	*Eucalyptus* sp.	China	HQ730837	HQ730863	HQ730850	MN263386
*Chrysofolia barringtoniae*	TBRC 5647	*Barringtonia acutangula*	Thailand	KU948046	KU948045	–	–
***Chrysofolia camelliae* sp. nov.**	**CGMCC3.27473 ***	***Camellia* sp.**	**China**	**PP658308**	**PP658399**	**PP665711**	**PP665721**
***Chrysofolia camelliae* sp. nov.**	**CGMCC3.27474**	***Camellia* sp.**	**China**	**PP658309**	**PP658400**	**PP665712**	**PP665722**
*Chrysofolia colombiana*	CPC 24986 *	*Eucalyptus urophylla* × *grandis*	Colombia	KR476738	KR476771	MN271829	–
*Chrysofolia coriariae*	GUCC 416.4 *	*Coriaria nepalensis*	China	OP581211	OP581237	OP688516	OP688542
*Chrysofolia coriariae*	GUCC 416.14	*Coriaria nepalensis*	China	OP581212	OP581238	OP688517	OP688543
*Chrysofolia fructicola*	GUCC 194121.1 *	*Rosa roxburghii*	China	ON791167	ON791210	ON815940	–
*Chrysofolia fructicola*	GUCC 194121.2	*Rosa roxburghii*	China	ON791168	ON791211	ON815941	–
*Chrysofolia galloides*	IFRDCC1022	*Rhus punjabensis* var. *sinica*	China	OR363211	OR363215	OR344509	OR344513
*Chrysofolia galloides*	IFRDCC1024 *	*Rhus punjabensis* var. *sinica*	China	OR363213	OR363217	OR344511	OR344515
*Chrysofolia kunmingensis*	KUNCC23-13350 *	*Coriaria nepalensis*	China	OR094460	OR094455	–	OR095639
*Chrysofolia kunmingensis*	KUNCC23-13351	*Coriaria nepalensis*	China	OR094461	OR094456	–	OR095640
*Chrysoporthe austroafricana*	CBS 112916	*Eucalyptus grandis*	South Africa	AF046892	–	–	AF273462
*Chrysoporthe cubensis*	CMW14394	*Eucalyptus grandis*	Cuba	DQ368773	–	–	AH015642
*Chrysoporthe deuterocubensis*	CMW8650	*Syzygium aromaticum*	Indonesia	AY084001	–	–	GQ290193
*Chrysoporthe doradensis*	CMW11286	*Eucalyptus deglupta*	Ecuador	JN942331	–	–	AY214254
*Chrysoporthe hodgesiana*	CMW10461	*Tibouchina semidecandra*	Colombia	AY692322	–	–	AY692325
*Chrysoporthe inopina*	CMW12727	*Tibouchina lepidota*	Colombia	DQ368777	–	–	AH015657
*Chrysoporthe syzygiicola*	CMW29940	*Syzygium guinense*	Zambia	FJ655005	MN172383	MN271839	FJ805236
*Chrysoporthe zambiensis*	CMW29929 *	*Eucalyptus grandis*	Zambia	JN942332	JN940846	–	–
*Dwiroopa punicae*	CBS 143163 *	*Punica granatum*	America	MK510676	MK510686	MH020056	MK510714
*Foliocryphia eucalypti*	CBS 124779 *	*Eucalyptus coccifera*	Australia	GQ303276	GQ303307	MN271861	JQ706128
*Foliocryphia eucalyptorum*	CBS 142536 *	*Eucalyptus* sp.	New Zealand	KY979772	KY979827	MN271862	KY979936

Notes: Newly generated sequences are in bold. The ex-type, ex-epitype, and ex-neotype strains are marked with *.

**Table 3 jof-10-00383-t003:** Species names, voucher or culture codes, hosts or substrate, locations, and corresponding GenBank accession numbers of DNA sequences used in the molecular phylogenetic analyses of *Dendrostoma*.

Species	Culture/Voucher	Host/Substrate	Locations	GenBank Accession Number			
				ITS	LSU	*tef1*	*rpb2*
*Dendrostoma atlanticum*	CBS 145804 *	*Castanea sativa*	Australia	MN447223	MN447223	MN432167	MN432160
*Dendrostoma atlanticum*	D303	*Castanea sativa*	France	MN447224	MN447224	MN432168	MN432161
*Dendrostoma aurorae*	CFCC 52753 *	*Castanea mollissima*	China	MH542498	MH542646	MH545447	MH545405
*Dendrostoma aurorae*	CFCC 52754	*Castanea mollissima*	China	MH542499	MH542647	MH545448	MH545406
*Dendrostoma castaneae*	CFCC 52745 *	*Castanea mollissima*	China	MH542488	MH542644	MH545437	MH545395
*Dendrostoma castaneae*	CFCC 52746	*Castanea mollissima*	China	MH542489	–	MH545438	MH545396
*Dendrostoma castaneicola*	CFCC 52743 *	*Castanea mollissima*	China	MH542496	–	MH545445	MH545403
*Dendrostoma castaneicola*	CFCC 52744	*Castanea mollissima*	China	MH542497	–	MH545446	MH545404
*Dendrostoma castaneum*	CBS 145803 *	*Castanea sativa*	Austria	MN447225	MN447225	MN432169	MN432162
*Dendrostoma castaneum*	D230	*Castanea sativa*	Italy	MN447226	MN447226	MN432170	–
***Dendrostoma castanopsidis* sp. nov.**	**CGMCC3.25675**	** *Castanopsis fargesii* **	**China**	**PP658310**	**PP658401**	**PP665713**	**PP665717**
***Dendrostoma castanopsidis* sp. nov.**	**CGMCC3.25676 ***	** *Castanopsis fargesii* **	**China**	**PP658311**	**PP658402**	**PP665714**	**PP665718**
*Dendrostoma chinense*	CFCC 52755 *	*Castanea mollissima*	China	MH542500	MH542648	MH545449	MH545407
*Dendrostoma chinense*	CFCC 52756	*Castanea mollissima*	China	MH542501	MH542649	MH545450	MH545408
*Dendrostoma covidicola*	GZCC 20–0355 *	*Fagus sylvatica*	China	MW261327	MW261325	MW262894	MW262892
*Dendrostoma covidicola*	MFLU 21–0001	*Fagus sylvatica*	China	MW261328	MW261326	–	MW262893
*Dendrostoma creticum*	CBS 145802 *	*Quercus coccifera*	Greece	MN447228	MN447228	MN432171	MN432163
*Dendrostoma dispersum*	CFCC 52730 *	*Quercus* sp.	China	MH542467	MH542629	MH545416	MH545374
*Dendrostoma dispersum*	CFCC 52731	*Quercus* sp.	China	MH542468	MH542630	MH545417	MH545375
*Dendrostoma donglingense*	CFCC 53148 *	*Quercus mongolica*	China	MN266206	MN265880	MN315480	MN315491
*Dendrostoma donglingense*	CFCC 53149	*Quercus mongolica*	China	MN266207	MN265881	MN315481	MN315492
*Dendrostoma elaeocarpi*	CFCC 53113 *	*Elaeocarpus decipiens*	China	MK432638	MK429908	MK578114	MK578096
*Dendrostoma elaeocarpi*	CFCC 53114	*Elaeocarpus decipiens*	China	MK432639	MK429909	MK578115	MK578097
*Dendrostoma istriacum*	CBS 145801 *	*Quercus ilex*	Croatia	MN447229	MN447229	MN432172	MN432164
*Dendrostoma leiphaemia*	CBS 187.37	*Quercus* sp.	–	MH855882	MH867393	–	–
*Dendrostoma luteum*	IMI 506898 *	*Castanea sativa*	England	MN648726	MN648728	MN812768	–
*Dendrostoma mali*	CFCC 52102 *	*Malus spectabilis*	China	MG682072	MG682012	MG682052	MG682032
*Dendrostoma osmanthi*	CFCC 52106 *	*Osmanthus fragrans*	China	MG682073	MG682013	MG682053	MG682033
*Dendrostoma osmanthi*	CFCC 52108	*Osmanthus fragrans*	China	MG682074	MG682014	MG682054	MG682034
*Dendrostoma parasiticum*	CFCC 52762 *	*Quercus wutaishanica*	China	MH542482	MH542638	MH545431	MH545389
*Dendrostoma parasiticum*	CFCC 52764	*Quercus aliena*	China	MH542483	MH542639	MH545432	MH545390
*Dendrostoma qinlingense*	CFCC 52732 *	*Quercus wutaishanica*	China	MH542471	MH542633	MH545420	MH545378
*Dendrostoma qinlingense*	CFCC 52733	*Quercus aliena*	China	MH542472	MH542634	MH545421	MH545379
*Dendrostoma quercinum*	CFCC 52103 *	*Quercus acutissima*	China	MG682077	MG682017	MG682057	MG682037
*Dendrostoma quercinum*	CFCC 52104	*Quercus acutissima*	China	MG682078	MG682018	MG682058	MG682038
*Dendrostoma quercus*	CFCC 52739 *	*Quercus* sp.	China	MH542476	MH542635	MH545425	MH545383
*Dendrostoma quercus*	CFCC 52738	*Quercus* sp.	China	MH542477	–	MH545426	MH545384
*Dendrostoma shaanxiense*	CFCC 52741 *	*Castanea mollissima*	China	MH542486	MH542642	MH545435	MH545393
*Dendrostoma shaanxiense*	CFCC 52742	*Castanea mollissima*	China	MH542487	MH542643	MH545436	MH545394
*Dendrostoma shandongense*	CFCC 52759 *	*Castanea mollissima*	China	MH542504	MH542652	MH545453	MH545411
*Dendrostoma shandongense*	CFCC 52760	*Castanea mollissima*	China	MH542505	MH542653	MH545454	MH545412
*Disculoides eucalypti*	CBS 132183 *	*Eucalyptus* sp.	Australia	JQ685517	JQ685523	MH545455	MH545413

Notes: Newly generated sequences are in bold. The ex-type, ex-epitype, and ex-neotype strains are marked with *.

**Table 4 jof-10-00383-t004:** Species names, voucher or culture codes, hosts or substrate, locations, and corresponding GenBank accession numbers of DNA sequences used in the molecular phylogenetic analyses of *Pseudoplagiostoma*.

Species	Culture/Voucher	Host/Substrate	Locations	GenBank Accession Number				
				ITS	LSU	*tef1*	*tub2*	*rpb2*
*Apoharknessia insueta*	CBS 111377 *	*Eucalyptus pellita*	Brazil	JQ706083	AY720814	MN271820	–	–
*Pseudoplagiostoma alsophilae*	SAUCC WZ0451 *	*Alsophila spinulosa*	China	OP810625	OP810631	OP828580	OP828586	OP828578
*Pseudoplagiostoma alsophilae*	SAUCC WZ0152	*Alsophila spinulosa*	China	OP810626	OP810632	OP828581	OP828587	OP828579
*Pseudoplagiostoma bambusae*	SAUCC 1206-4 *	*Bambusoideae* sp.	China	OP810629	OP810635	OP828584	OP828590	–
*Pseudoplagiostoma bambusae*	SAUCC 1206-6	*Bambusoideae* sp.	China	OP810630	OP810636	OP828585	OP828591	–
*Pseudoplagiostoma corymbiae*	CBS 132529 *	*Corymbia* sp.	Australia	JX069861	JX069845	–	–	–
*Pseudoplagiostoma corymbiicola*	CBS 145052 *	*Corymbia citriodora*	Australia	MK047425	MK047476	MK047558	MK047577	–
*Pseudoplagiostoma dipterocarpi*	TBRC 1895 *	*Dipterocarpus tuberculatus*	Thailand	KR994682	KR994683	–	–	–
*Pseudoplagiostoma dipterocarpicola*	MFLUCC 21-0142 *	*Dipterocarpus* sp.	Thailand	OM228844	OM228842	OM219629	OM219638	–
*Pseudoplagiostoma dipterocarpicola*	MFLUCC 21-0114	*Dipterocarpus* sp.	Thailand	OM228843	OM228841	OM219628	OM219637	–
*Pseudoplagiostoma eucalypti*	CBS 124807 *	*Eucalyptus urophylla*	Venezuela	GU973512	GU973606	GU973542	GU973575	–
*Pseudoplagiostoma eucalypti*	CPC 14161	*Eucalyptus camaldulensis*	Viet Nam	GU973510	GU973604	GU973540	GU973573	–
*Pseudoplagiostoma machili*	SAUCC BW0233 *	*Machilus nanmu*	China	OP810627	OP810633	OP828582	OP828588	–
*Pseudoplagiostoma machili*	SAUCC BW0221	*Machilus nanmu*	China	OP810628	OP810634	OP828583	OP828589	–
*Pseudoplagiostoma mangiferae*	KUMCC 180-179 *	*Mangifera* sp.	China	MK084824	MK084825	–	–	–
*Pseudoplagiostoma myracrodruonis*	URM 7799 *	*Astronium urundeuva*	Brazil	MG870421	MK982151	MK982557	MN019566	MK977723
*Pseudoplagiostoma myracrodruonis*	URM 8123	*Astronium urundeuva*	Brazil	MK982150	MK982152	MK982558	MN019567	MK977724
*Pseudoplagiostoma oldii*	CBS 115722	*Eucalyptus camaldulensis*	Australia	GU973535	GU973610	GU973565	GU993864	–
*Pseudoplagiostoma oldii*	CBS 124808 *	*Eucalyptus camaldulensis*	Australia	GU973534	GU973609	GU973564	GU993862	–
*Pseudoplagiostoma variabile*	CBS 113067 *	*Eucalyptus globulus*	Uruguay	GU973536	GU973611	GU973566	GU993863	–
***Pseudoplagiostoma wuyishanense* sp. nov.**	**CGMCC3.25367**	**on dead branches**	**China**	**PP658312**	**PP658403**	**PP665715**	**PP665723**	**PP665719**
***Pseudoplagiostoma wuyishanense* sp. nov.**	**CGMCC3.25368 ***	**on dead branches**	**China**	**PP658313**	**PP658404**	**PP665716**	**PP665724**	**PP665720**

Notes: Newly generated sequences are in bold. The ex-type, ex-epitype, and ex-neotype strains are marked with *.

## Data Availability

All sequences generated in this study were submitted to the NCBI database.

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
