# Peer review of "Phylogenetic and Morphological Evidence for Three New Species of Diaporthales (Ascomycota) from Fujian Province, China"

_jof, 2024, doi:10.3390/jof10060383_

Round 1

Reviewer 1 Report

The manuscript is understandable and clear. Some words require grammatical corrections. However, the authors should review the phylogenetic analyses of the Chrysofolia genus and include other recently published species. The pictures are lovely, but Figure 4 (g-h) is not clear. 

Abstract Line 25

The authors should, in the abstract, explain which plants were chosen or at least the families.

Keywords:

Diaporthales is already in the title, change to, for instance, phytopathogenic fungi.

Line 67

Chrysofolia includes five species in the Index Fungorum:

Chrysofolia barringtoniae 
Chrysofolia colombiana 
Chrysofolia coriariae 
Chrysofolia fructicola 
Chrysofolia kunmingensis 

Also, the authors should not rely only on  this plataform because some data are not updated. For example, Ma T, Yang Z.  et al. (2023) described a new species, Chrysofolia galloides T. Ma & Z. X. Yang, sp. nov. (Mycobank: MB850312), and this táxon is not include in the index fungorum

First report of a new species, Chrysofolia galloides sp.nov., causing black spots on gallnuts of Rhus punjabensis var. sinica in China. Plant Dis. 2023 Nov 6. doi: 10.1094/PDIS-08-23-1548-PDN. Epub ahead of print. PMID: 37933147.

Results

Chrysofolia galloides sistered to Chrysofolia coriariae. The authors should include the sequence of the species to see the agroupment. Genbank accession OR363211.

Line 192

*Bayesian posterior probabilities*

Line 211

*Bayesian inference*

Taxonomy

Line 244

Indicate with an arrow the condigeneous cells in Figure 4 (g,h). As the picture is not clear, change it for a better one.

Line 246

Chrysofolia on

Line 266

(e-g) conidiogenous cells with conidia.

278

Fawn by age

285

Description

Line 305

117º 39’ 23’’ E, 27º 40’ 13’’ ???

Line 310

Brown

Line 317

Grey by age

Line 329

117º 39’ 23’’ E, 27º 40’ 13’’ ???

Line 370

The authors should update the paragraph that starts in line 370.

Line 412

warranted.

Author Response

Dear Editors and Reviewers:

Thank you for your letter and comments relating to our manuscript entitled “Phylogenetic and morphological evidence for three new species of Diaporthales from Fujian Province of China” (ID: jof- 3016089). The comments were very helpful in revising and improving our manuscript as well as emphasizing the significance to our research. We have read the comments carefully and made corrections accordingly. Revised portions are marked in blue in the manuscript. The main corrections in the paper and our responses to the reviewer’s comments are given below. We hope that the revisions in the manuscript and our accompanying responses will be sufficient to make our manuscript suitable for publication in the Journal of Fungi.

Responses to the comments of the reviewer:

Reviewer 1#

Comments 1: The manuscript is understandable and clear. Some words require grammatical corrections. However, the authors should review the phylogenetic analyses of the Chrysofolia genus and include other recently published species. The pictures are lovely, but Figure 4 (g-h) is not clear. 

Response 1: We have analyzed the Chrysofolia genus and other recently published species in the revision, and made Figure 4 (g-h) clear.

Comments 2: Abstract Line 25

The authors should, in the abstract, explain which plants were chosen or at least the families.

Response 2: We have provided information of plant hosts and chosen plants in the abstract.

Fungal samples were obtained from dead branches and diseased leaves of Camellia (Theaceae) and Castanopsis (Fagaceae) in Fujian Province, China.

Comments 3: Keywords:

Diaporthales is already in the title, change to, for instance, phytopathogenic fungi.

Response 3: The keywords section has been edited.

Keywords: multigene phylogeny; phytopathogenic fungi; Pseudoplagiostomataceae; taxonomy

Comments 4: Line 67, Chrysofolia includes five species in the Index Fungorum:

Chrysofolia barringtoniae, Chrysofolia colombiana, Chrysofolia coriariae, Chrysofolia fructicola, Chrysofolia kunmingensis 

Also, the authors should not rely only on this plataform because some data are not updated. For example, Ma T, Yang Z.  et al. (2023) described a new species, Chrysofolia galloides T. Ma & Z. X. Yang, sp. nov. (Mycobank: MB850312), and this táxon is not include in the index fungorum.

First report of a new species, Chrysofolia galloides sp.nov., causing black spots on gallnuts of Rhus punjabensis var. sinica in China. Plant Dis. 2023 Nov 6. doi: 10.1094/PDIS-08-23-1548-PDN. Epub ahead of print. PMID: 37933147.

Response 4: We have constructed a new phylogenetic tree includes Chrysofolia barringtoniae, Chrysofolia colombiana, Chrysofolia coriariae, Chrysofolia fructicola, Chrysofolia galloides and Chrysofolia kunmingensis. The new phylogenetic tree doesn't influence our previous result. Thank you for your valuable comments.

Comments 5: Results, Chrysofolia galloides sistered to Chrysofolia coriariae. The authors should include the sequence of the species to see the agroupment. Genbank accession OR363211.

Response 5: We have constructed a new phylogenetic tree includes six species of Chrysofolia based on published papers and provided the latest GenBank accession numbers (Table 2).

Comments 6: Line 192, *Bayesian posterior probabilities*; Line 211, *Bayesian inference*; Taxonomy, Line 244

Response 6: We have added spacing where it was missing.

Comments 7: Indicate with an arrow the condigeneous cells in Figure 4 (g,h). As the picture is not clear, change it for a better one.

Response 7: We have made pictures clearer and added arrows to indicate.

Comments 8: Line 246, Chrysofolia on; Line 266, (e-g) conidiogenous cells with conidia; Line 278, Fawn by age; Line 285, Description; Line 305, 117º 39’ 23’’ E, 27º 40’ 13’’ ???; Line 310, Brown; Line 317, Grey by age; Line 329, 117º 39’ 23’’ E, 27º 40’ 13’’ ???; Line 370, The authors should update the paragraph that starts in line 370.;  Line 412, warranted.

Response 8: We have rechecked the errors throughout the whole manuscript and added spacing where it was missing. We have updated the paragraph that starts in line 370. Presently, six species of Chrysofolia are accepted and have been reported as plant pathogens and endophytes. Chrysofolia barringtoniae, Chrysofolia fructicola and Chrysofolia kunmingensis were isolated as endophytes from leaves of Barringtonia acutangula, Rosa roxburghii and Coriaria nepalensis, respectively, whereas Chrysofolia coriariae appears to be a pathogen of the Coriaria nepalensis plant.

We tried our best to improve the manuscript and made some changes marked in blue in revised paper which will not influence the content and framework of the paper. We appreciate for Editors/Reviewers’ warm work earnestly and hope the revision will meet with your approval. Once again, thank you very much for your comments and suggestions.

Kind regards,

Junzhi Qiu

E-mail address: [email protected]

Reviewer 2 Report

A review on a manuscript “ Phylogenetic and morphological evidence for three new species of Diaporthales from Fujian Province of China” written by Taichang Mu et al.

In my opinion, this is an excellent mycological contribution. All three newly introduced species are well supported phenotypically and phylogenetically, accompanied with well diagnostic description and illustrations.

I have only several points here:

1 Title  to add after order  …. Diaporthales (Ascomycota)….  OPTIONAL

Line 39  it is “ of about 32”  should be  “ of 32 “

Line 49 replace [6-8] into end of the sentence into Line 50.

Line 173 it is “ andsecond”

Line 200 it is typo in “construction”

Into all dimension, please do round off numbers e.g. Line 230 it is “3.7 – 6.1”  should be 4.0 – 6.0 µm (due to optical microscope 0.5 µm resolution, regardless of  a computer software measurement tool) etc. all in the text. REQUESTED

Line 246 it is “ Chrysofoliaon”

Line 299  use “narrower” as the comparative form of “narrow.”

Line 332  branches in China.   ( what kind of branches, please insert into accordingly)

Line 355 add        ……. , respectively) in Diaporthales……

Line 371  it is “ endopytes”

Figure 6 is really fantastic! And I do and for sure readers will appreciate the excellent graphics of the Figure 2 (phylogram with very smart depiction of geographic origin and hosts, excellent!)

---- end---

as above 

Author Response

Dear Editors and Reviewers:

Thank you for your letter and comments relating to our manuscript entitled “Phylogenetic and morphological evidence for three new species of Diaporthales from Fujian Province of China” (ID: jof- 3016089). The comments were very helpful in revising and improving our manuscript as well as emphasizing the significance to our research. We have read the comments carefully and made corrections accordingly. Revised portions are marked in blue in the manuscript. The main corrections in the paper and our responses to the reviewer’s comments are given below. We hope that the revisions in the manuscript and our accompanying responses will be sufficient to make our manuscript suitable for publication in the Journal of Fungi.

Responses to the comments of the reviewer:

Reviewer 2#

Comments 1: Title to add after order  …. Diaporthales (Ascomycota)….  OPTIONAL

Response 1: We have revised it.

Phylogenetic and morphological evidence for three new species of Diaporthales (Ascomycota) from Fujian Province of China

Comments 2: Line 39,  it is “ of about 32”  should be  “ of 32 “

Response 2: We have revised it.

Comments 3: Line 49 replace [6-8] into end of the sentence into Line 50.

Response 3: We have revised it.

Comments 4: Line 173, it is “and second”

Response 4: We have added spacing where it was missing.

Comments 5: Line 200, it is typo in “construction”

Response 5: We have revised it.

Comments 6: Into all dimension, please do round off numbers e.g. Line 230 it is “3.7 – 6.1”  should be 4.0 – 6.0 µm (due to optical microscope 0.5 µm resolution, regardless of  a computer software measurement tool) etc. all in the text. REQUESTED

Response 6: We have revised it.

Comments 7: Line 246, it is “ Chrysofoliaon”

Response 7: We have added spacing where it was missing.

Comments 8: Line 299, use “narrower” as the comparative form of “narrow.”

Response 8: We have revised it.

Comments 9: Line 332,  branches in China.   (what kind of branches, please insert into accordingly)

Response 9: We have revised it.

In this study, Pseudoplagiostoma wuyishanense sp. nov. is described from dead branches of an unidentified broadleaf tree in China.

Comments 10: Line 355, add        ……. , respectively) in Diaporthales……

Response 10: We have revised it.

Comments 11: Line 371,  it is “ endopytes”

Response 11: We have revised it.

Comments 12: Figure 6 is really fantastic! And I do and for sure readers will appreciate the excellent graphics of the Figure 2 (phylogram with very smart depiction of geographic origin and hosts, excellent!)

Response 12: Thank you for your wonderful comments. We have revised it.

We tried our best to improve the manuscript and made some changes marked in blue in revised paper which will not influence the content and framework of the paper. We appreciate for Editors/Reviewers’ warm work earnestly and hope the revision will meet with your approval. Once again, thank you very much for your comments and suggestions.

Kind regards,

Junzhi Qiu

E-mail address: [email protected]
